



# A CO$_2$-free cloud mask from IASI radiances for climate applications

Simon Whitburn[1], Lieven Clarisse[1], Marc Crapeau[2], Thomas August[2], Tim Hultberg[2], Pierre François Coheur[1], and Cathy Clerbaux[1,3]

[1]Spectroscopy, Quantum Chemistry and Atmospheric Remote Sensing (SQUARES), Université libre de Bruxelles (ULB), Brussels, Belgium
[2]European Organisation for the Exploitation of Meteorological Satellites (EUMETSAT), Darmstadt, Germany
[3]LATMOS/IPSL, Sorbonne Université, UVSQ, CNRS, Paris, France

**Correspondence:** Simon Whitburn (simon.whitburn@ulb.be)

**Abstract.** With more than 15 years of continuous and consistent measurements, the Infrared Atmospheric Sounding Interferometer (IASI) radiance dataset is becoming a reference climate data record. To be exploited to its full potential, it requires a cloud filter that is both accurate, unbiased over the full IASI lifespan, and strict enough to be used in satellite data retrieval schemes. Here, we present a new cloud detection algorithm which combines (1) a high sensitivity, (2) a good consistency over

the whole IASI time series and between the different copies of the instrument flying on board the suite of Metop satellites and (3) simplicity in its parametrization. The method is based on a supervised neural network (NN) and relies, as input parameters, on the IASI radiance measurements only. The robustness of the cloud mask over time is ensured in particular by avoiding the IASI channels that are influenced by CO$_2$, N$_2$O, CH$_4$, CFC-11 and CFC-12 absorption lines and those corresponding to the $\nu_2$ H$_2$O absorption band. As a reference dataset for the training, the latest version of the operational IASI Level 2 (L2)

cloud product is used. We provide different illustrations of the NN cloud product, including comparisons with other existing products. We find a very good agreement overall with the last version of the operational IASI L2 with an identical mean annual cloud amount and a pixel-by-pixel correspondence of about 87%. The comparison with the other cloud products shows a good correspondence in the main cloud regimes but with sometimes large differences in the mean cloud amount (up to 10%) due to the specificities of each of the different products. We also show the good capability of the NN product to differentiate clouds

from dust plumes.

## 1   Introduction

Clouds cover between 70% to 80% of the Earth's surface, at any moment (Lavanant et al., 2011; Stubenrauch et al., 2017). Because of their importance on the weather, the water cycle and the Earth radiation budget, the development of long accurate and coherent time series of cloud properties (e.g. cloud amount, cloud top height, optical thickness, cloud type) is essential

for improving our understanding of the climate, its past and future evolution. We can rely for this on satellite observations which allow to study the daily cloud coverage at global scales. Their use in the detection of clouds and to derive climatologies has begun in the 1980s. One of the first global cloud climate data record is the International Satellite Cloud Climatology (ISCCP) which started in 1982 (Schiffer and Rossow, 1983; Rossow and Schiffer, 1999). With more than 40 years of record,





it has become today a reference for climate analysis. Since then, the measurements from a variety of sounders on board

polar and geostationnary platforms have been used to detect and characterize clouds (e.g. Kaspar et al., 2009; Karlsson et al., 2013, 2017; Stengel et al., 2017; Feofilov and Stubenrauch, 2017). Despite this, they remain today one of the largest source of uncertainties in future climate projections (Schneider et al., 2017; Zelinka et al., 2017; Satoh et al., 2018). Besides their importance for modeling the Earth's climate, accurate detection of cloud-free (clear) scenes from satellite measurements is also an essential preprocessing step for most climate and atmospheric satellite applications. This is especially the case in most

trace gas (e.g. Van Damme et al., 2017) and dust retrieval schemes (Clarisse et al., 2019) or to derive the Earth Outgoing Longwave Radiation (OLR) budget (Whitburn et al., 2020), as the presence of even small cloud amounts in the instrument field-of-view will significantly impact the radiance at the top of the atmosphere.

  Here, we present a new cloud detection algorithm for the measurements of the Infrared Atmospheric Sounding Interferometer (IASI) flying on board the suite of Metop satellites (Metop-A, -B and -C) since 2007 (Clerbaux et al., 2009; Hilton et al., 2012).

IASI is a hyperspectral infrared sounder which covers a large part of the thermal infrared region (between 645 and 2760 $cm^{-1}$) without any gaps and with a uniform spectral sampling of 0.25 $cm^{-1}$. It provides a quasi global coverage of the Earth twice daily (at 09.30 AM and PM) with a relatively small elliptical footprint on the ground varying from 12 km × 12 km (at nadir) up to 20 km × 39 km (off nadir). These characteristics combined with the good stability of the instrument over more than 15 years (Saunders et al., 2021) and the consistency in the measurements between the three instruments (Bouillon et al., 2020)

makes the IASI radiance dataset an excellent fundamental climate data record. For the detection and the characterization of the clouds in the field of observation of IASI, several products already exist: (i) the IASI-CIRS (Clouds from Infrared Sounders) cloud product (Feofilov and Stubenrauch, 2017; Stubenrauch et al., 2017), (ii) the integrated cloud fraction from the AVHRR (Advanced Very High Resolution Radiometer) imager, flying with IASI (product identifier GEUMAvhrr1BCldFrac, Guidard et al., 2011) and (iii) the EUMETSAT's operationally distributed IASI Level 2 (L2) cloud product (August et al., 2012). While

these three products are generally of very good quality, they also suffer from weaknesses in the identification of the clear-sky scenes.

  The IASI-CIRS cloud product is derived from an operational and modular cloud retrieval algorithm suite (CIRS) developed and maintained at the Laboratoire de Météorologie Dynamique (LMD) (Feofilov and Stubenrauch, 2017; Stubenrauch et al., 2017). The retrieval relies on a weighted $\chi^2$ method using channels around the 15 $\mu m$ $CO_2$ absorption band (Stubenrauch et al.,

1999). It derives cloud pressure and effective emissivity which are then used to assign a cloud type (8 classes in total, including clear-sky) (Feofilov and Stubenrauch, 2017; Stubenrauch et al., 2017). So far, the algorithm has been applied to the High-resolution Infrared Radiation Sounder (HIRS), the Atmospheric Infrared Sounder (AIRS) and the IASI dataset (Feofilov and Stubenrauch, 2017; Stubenrauch et al., 2017). The dataset is consistent over time, in particular because of the consideration of the increasing $CO_2$ atmospheric concentrations in the calculation to avoid long-term biases. However, the product appears not

to be strict enough for using as cloud removal preprocessing phase in satellite data retrieval schemes of geophysical variables, as it was never designed for this purpose.

  The AVHRR instrument onboard the Metop satellites measures the radiation in five spectral channels in both visible and infrared bands with a spatial resolution of 1.1 km at nadir. The integrated AVHRR cloud fraction (hereafter abbreviated L1C-





AVHRR), which presents the advantage of being directly embedded in the IASI L1C products, represents the percentage of

AVHRR cloudy pixels in the IASI field of view (Guidard et al., 2011; Farouk et al., 2019). While the product performs relatively well in the tropical and mid-latitude regions, the sensitivity to cloud detection decreases at high latitudes, especially during the winter period likely due to the absence of visible light and the very cold surface temperatures, as well as in conditions of high albedo (e.g. snow and ice).

Finally, the official IASI L2 cloud product is part of the operational processing chain of the L2 IASI Product Processing

Facility (PPF) operationally disseminated by EUMETSAT (European organization for the exploitation of METeorological SATellites). It includes a cloud flag and also provides, for each cloud scene, the cloud top pressure, the cloud fraction and the cloud phase (August et al., 2012; IASI Level 2, 2017). Since it was first deployed in 2007, the L2 meteorological data have undergone a series of updates (summarized up to the version 6.4 in Table 2 in Bouillon et al. (2020)) improving their quality to progressively reach excellent performances today. Until the version 6.4, the cloud detection was based on three

independent tests: an AVHRR collocated cloud mask, a classical window channel test (observed minus simulated spectrum) based on numerical weather predictions (NWP) and a neural network applied to the IASI and AVHRR measurements. A scene was flagged as clear if all tests concluded to the absence of clouds. For all IASI pixels declared as cloudy, the characterization was then performed using a $CO_2$-slicing and a $\chi^2$ method (August et al., 2012; IASI Level 2, 2017). Since the version 6.5 released in December 2019, a new cloud detection scheme has been introduced. It is based on a IASI-only optimal estimation

algorithm that retrieve the cloud fraction and the cloud top pressure. The distinction between cloudy and clear-sky scenes is based on the retrieved cloud fraction. The same algorithm is planned to be used for the measurements of the future IASI-NG (New Generation) (Crevoisier et al., 2014) and MTG-IRS (Meteosat third generation) sounders. The variational cloud retrieval algorithm is described in MTG-IRS Level 2 ATBD (https://www.eumetsat.int/media/45439). In the IASI case, a check on the field of view inhomogeneity, based on the collocated AVHRR radiances, is used to identify some additional clouds.

An entire reprocessed cloud series with the latest version of the operational L2 cloud product (version 6.5) has, however, not been released yet, leading to discontinuities in the current data record which makes it less suitable for use in long-term studies. Moreover, a positive trend is also observed in the time series of the cloud amount related to the $CO_2$ increase in the atmosphere, as will be shown in Sect. 4.2. Lastly, another issue that was reported, at least for the earlier versions of the product, is the presence of false cloud detections in particular in the center of large dust plumes (Clarisse et al., 2019).

The limitations of the existing products have triggered the development of a sensitive and coherent IASI cloud detection dataset to be used for climate research and for studying trends of atmospheric compounds. The algorithm is based on the use of a Neural Network (NN) and relies on IASI radiance information only, as was also successfully achieved recently to retrieve surface skin temperatures (Safieddine et al., 2020). In the next section we detail the neural network retrieval algorithm (setup, parametrization, training). IASI-NN derived average cloud distributions and cloud seasonality are presented in Sect. 3.

Section 4 gives a first assessment of the cloud product and an inter-comparison with other existing cloud products and cloud climate data records. We show that the IASI cloud mask presented here is both accurate and consistent over time and between the different IASI instruments. Conclusions are given in Sect. 5.





**Table 1.** List of IASI wavenumbers selected for the training of the cloud detection neural network.

| IASI wavenumber (cm$^{-1}$) |
|---|
| 826.00, 827.50, 861.50, 865.50, 866.25, 869.25, 871.25, 874.75, 877.00, 878.50, 880.00, 883.75, 885.75, 887.25, 891.25, 894.25, 897.75, 899.75, 901.50, 902.50, 905.00, 994.50, 996.25, 999.50, 1001.50, 1004.75, 1006.50, 1009.75, 1011.50, 1014.50, 2145.50, 2147.25, 2150.00, 2152.50, 2153.50, 2157.25, 2158.25, 2161.75, 2164.75, 2166.75, 2169.25, 2172.75, 2174.25, 2176.25, 2177.75 |

## 2 The neural network: setup, training, retrieval and postfiltering

The goal of the proposed algorithm is to produce a sensitive and consistent (unbiased) cloud mask over the entire IASI lifespan

using as a reference dataset the last version (v6.5) of the operational IASI L2 cloud product (August et al., 2012). The method is based on a supervised NN relying on the IASI radiance spectra only, as opposed to the IASI-L2 which also relies on the measurements of other instruments and model simulation. The consistency over time is ensured by careful selection of the IASI channels considered as input in the NN. In particular, spectral regions affected by the $CO_2$, $N_2O$, $CH_4$, CFC-11 and CFC-12 absorption lines were excluded from the selection procedure as their concentration are evolving steadily in the atmosphere and

were found to affect significantly the spectral OLR over 10 years of IASI measurements (De Longueville et al., 2021; Whitburn et al., 2021). Other long-lived species whose concentrations are changing in the atmosphere also exist but are not expected to introduce a bias in the cloud detection. Similarly, we also removed the region corresponding to the $\nu_2$ $H_2O$ absorption band (between 1050 and 2140 cm$^{-1}$) as preliminary tests indicated spurious trends related to the activity of major climate phenomena (e.g. El Niño Southern Oscillation). Finally, the portion of the spectrum affected by the solar reflectance (above 2400 cm$^{-1}$)

was avoided as well as it would introduce inconsistencies between day and night measurements.

In the remaining spectral regions (see Fig. 1), we selected, from a mean IASI spectrum, channels associated with minimum and maximum brightness temperatures. Indeed, the maxima represent mainly background radiation (coming from either clouds or the surface or both). The minima correspond to channels strongly affected by trace gas absorptions. The difference between the maxima and the minima provide information on cloud opacity. By selecting spectral regions affected respectively by $H_2O$,

$O_3$ and CO the network is sufficiently robust against local changes in their concentration (e.g. at high latitude the water vapour column is low, but in that case the $O_3$ channels are still available). In practice, the spectral regions were evaluated in groups of 20 channels from which the minimum and the maximum brightness temperature was selected. An exclusion criterion was applied on neighboring channels (two minima/maxima must be at least three channels apart) to reduce redundancy and ease training of the network (as neighbouring IASI channels are highly correlated). We ended-up with 45 channels mainly located in the

atmospheric window 1 and 3 and on the low wavenumber part of the $\nu_3$ $O_3$ absorption band. The wavenumbers corresponding to the selected IASI channels are given in Table 1 and are shown in Fig. 1.

To get the training representative of real input data, the training set should be as large and as extensive as possible while remaining reasonable in terms of computational ressources required. For this, about 54.000 clear and 54.000 cloud scenes derived from Metop-B observations were taken randomly over one year (2020). Since the EUMETSAT IASI-L2 product





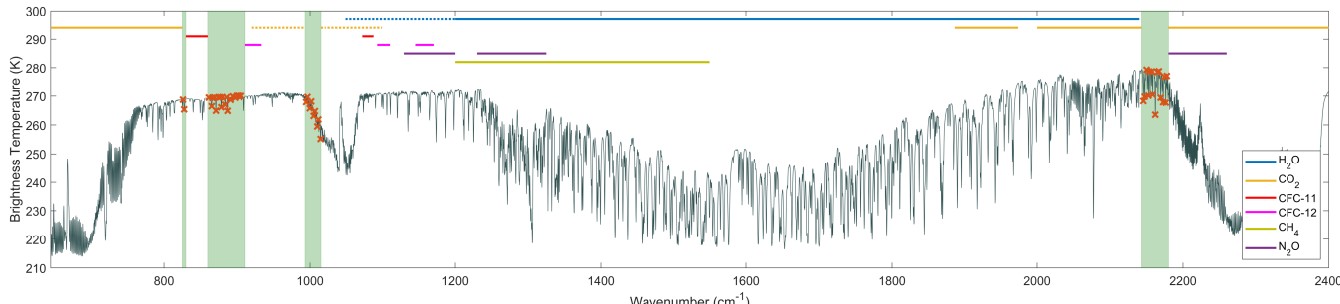

**Figure 1.** Mean IASI spectrum in brightness temperature with the 45 channels selected for the training displayed (red crosses). The green shaded area corresponds to the regions considered for the channels selection. The colored lines on the top represent the regions of sensitivity to $H_2O$, $CO_2$, $CH_4$, $N_2O$, CFC-11 and CFC-12 excluded.

provides a value of cloud fraction in the IASI pixel, we first converted it into a cloud flag: a scene was considered as cloud free only if the cloud fraction was strictly equal to zero. With these and with the corresponding IASI radiance at the 45 selected channels, the NN was trained. The NN consists of a supervised two-layer pattern recognition network (Kothari and Oh, 1993) with one sigmoid hidden layer of 20 neurons and one output layer of one linear neuron. The training algorithm is a Levenberg-Marquardt backpropagation. For the training, the data set was divided randomly into three parts: (1) the training set (95%), (2) a validation set (4%) and a test set (1%). In addition to the 45 channels, we also included the surface elevation as input parameter for the NN taken from the "National Geophysical Data Center TerrainBase Global DTM Version 1.0" (ftp://ftp.ngdc.noaa.gov/Solid_Earth/cdroms/TerrainBase_94/). In total, we performed 10 different trainings and we selected the least affected with dust (see Sect. 4.3). The performance of the selected training reaches 87.3% with an equivalent number of misdetections in the clear and cloud group.

The actual retrieval provides a value comprised between 0 and 1. If the average cloud fraction was about 50% on Earth, the threshold for separating the clear and cloud scenes would be equal to 0.5 as the training was performed with the same amount of data in the two groups. However in practice, as mentioned in the introduction, cloud contaminated scenes dominate and the optimal threshold (minimizing the differences with the L2) has to be determined. For this, we calculated the mean NN cloud amount for the year 2020 in a $0.25° \times 0.25°$ grid (calculated as the number of the scenes flagged as cloud over the total number of observations) by considering a set of different thresholds and found the one which minimize the difference with the L2-derived cloud amount. A separate threshold, of respectively 0.175 and 0.275, was defined for land and for sea measurements. As we demonstrate in Sect. 4.1, this threshold can be adjusted depending on the application.

After application of the network and the threshold to derive the cloud mask, a number of low temperature observations stood out as being clearly misclassified, having a very low temperature (<270 K) at midlatitude (mostly over ocean). While the number of such observations is very low (<0.2%), they were important enough to be removed with a postfilter which was constructed as follows. We built a monthly climatology of the mean and standard deviation of the brightness temperature (BT) calculated for one channel at 821.75 cm$^{-1}$ on a $1° \times 1°$ grid from the 2008–2020 data. The channel was carefully selected





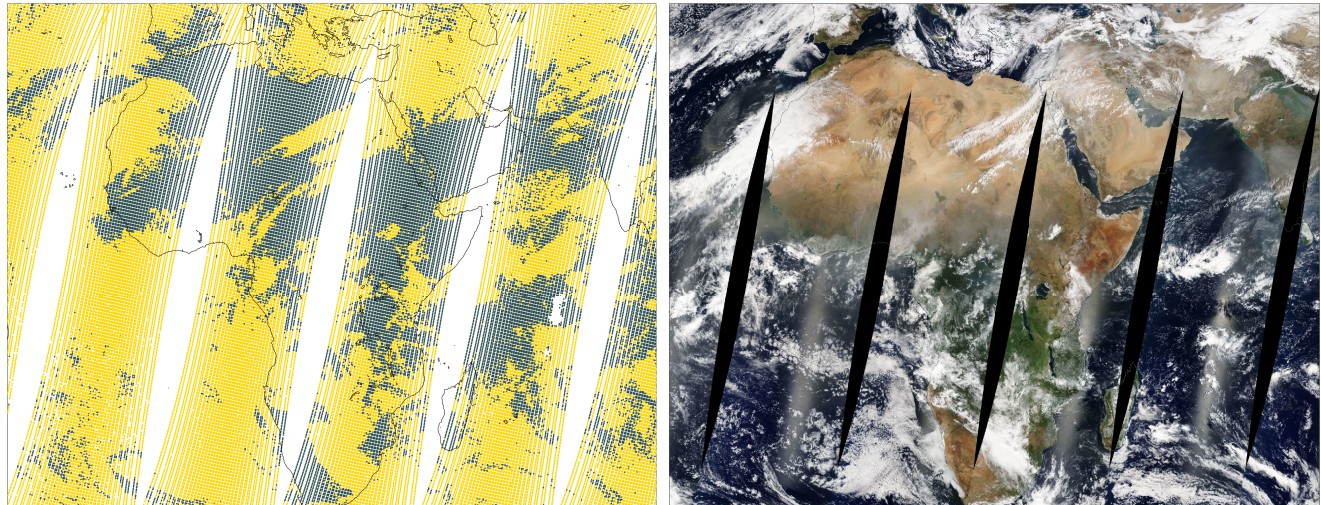

**Figure 2.** Left: Example of cloud detection from the IASI neural network algorithm for one day of measurements (2018/02/15, morning overpass). Cloud scenes are in yellow, clear-sky scenes in blue. Right: MODIS Terra corrected reflectance (true colors) imagery for the same day (from NASA Worldview).

to avoid the regions of absorption of the main atmospheric absorbers and the wavenumbers strongly affected by the surface type (water, snow, ice, deserts, ...). Any retrieval initially declared as clear is flagged as cloudy if the BT associated to the measurement is lower than the mean BT minus 3 times its standard deviation for the corresponding month in the $1° \times 1°$ grid.

An example of retrieved cloud mask on 15 February 2018 for the morning overpasses of IASI over Africa, the Arabian Peninsula and the western part of the Indian Ocean is given in Fig. 2. Yellow points correspond to pixels flagged as cloudy. The MODIS Terra corrected reflectance imagery for the same day is shown as well. An excellent correspondence is found, with both high opaque clouds and thin cirrus detected. A good distinction is also obtained between clouds and dust plumes as observed on the west coast of Africa north of the equator. The sensitivity to dust is further assessed in Sect. 4.3.

## 3 Average distributions

The entire IASI time series has been processed for Metop-A,-B and -C. The computational time is fast, taking about 2.5 minutes in CPU time per day processed. For Metop-A before 2017, a reprocessed radiance dataset with the latest version of the L1C has been released by EUMETSAT in 2018 (EUMETSAT, 2018). This allows us to produce a consistent time series of the cloud mask for the 15 years of IASI observations. In this section, we give a short overview of the large-scale cloud regimes and seasonality as captured by the IASI-NN cloud product. The mean cloud amount (%, also referred indifferently to as cloud cover or cloud fraction from here onwards) derived from Metop-A observations separately from the morning and the evening overpasses between 2008 and 2020 in a $0.25° \times 0.25°$ is shown on Fig. 3. It is calculated as the number of observations flagged





as cloudy over the total number of observations. The two distributions are very similar with, on average, differences in the mean
cloud amount lower than 2% for seas and 1% for land.

The large-scale patterns are very similar to those reported in previous studies (e.g. Wylie D. P., 1999; King et al., 2013).
In general, clear-sky is more common over land than over oceans: globally, about 85% and 57% of all observations over seas
and land, respectively, are flagged as cloudy. The largest cloud amounts are found at mid- to high latitudes over oceans where
cold air condenses water vapor into clouds. This is especially the case in the Southern Hemisphere (below about 45° S), where
the cloud amount exceeds 95% on average over the year. In the Northern Hemisphere, the North Atlantic and North Pacific
(from 40° N) exhibits similar values. Close to the equator, a band of high cloud cover (>90%) located at the Intertropical
Convergence Zone (ITCZ) corresponding to the region of convergence of the northeast and the southeast trade winds is also
clearly visible. Two other regions characterized by a very high mean cloud cover are located along the coasts of Peru and Chile
and in the Atlantic, west of Angola. Conversely, lower cloud amounts of about 50 to 70% are observed over the subtropical
gyres of the oceans.

Above land, the cloud distribution is more variable with desert areas characterized by a quasi absence of cloud over the
whole year (typically, around 20% for the Sahara and the Arabian Peninsula) while intertropical regions show a cloud cover of
about 80–85%. Mid- and high latitudes in the Northern Hemisphere, for their parts, are typically characterized by a mean cloud
fraction of 70–75%. Desert conditions with about 30% mean cloud cover are also observed in the Eastern part of Antarctica
characterized by high altitudes reaching 4 km on the Antarctic Plateau (Listowski et al., 2019). Over high latitudes in the
Northern and the Southern Hemisphere globally (between 60° and 90° N and S, for land and seas together), we find a mean
cloud coverage of about 76% and 66%, respectively. While for the Northern Hemisphere, this is in excellent agreement with
values reported from the Cloud-Aerosol LiDAR and Infrared Pathfinder Satellite Observation (CALIPSO) in Karlsson and
Devasthale (2018), the cloud coverage in the Southern Hemisphere might be slightly underestimated by about 10% (around
76% from CALIPSO (Karlsson and Devasthale, 2018)).

Note the presence of high cloud cover values observed over the tall mountain ranges in the different regions of the world (e.g.
Himalayas, Ural mountains, Andes, Rocky mountains). These are likely exaggerated by false cloud detection due to the climatic
conditions in these areas (lower thermal contrast between the surface and the cloud, drier air, ...) or to the heterogeneity of the
topography within the IASI pixel which make the distinction between clear and cloudy pixels more difficult. These patterns are
also observed in the distributions derived from other cloud products (see Sect. 4). In addition, for the nighttime distribution,
patches of slightly higher cloud coverage are observed over deserts, in particular for the Sahara desert, which probably reflect
emissivity features associated with a lower sensitivity.

When looking at the seasonal average derived over the 2008-2020 period from the daytime measurements (Fig. 4), we find
strong variations in agreement with the known seasonal cycles of the clouds around the globe. Close to the equator, the band
of high cloud amount moves progressively in latitude with the ITCZ from its Northernmost position around 10° N during
the boreal summer (Fig. 4c, June–August) to its Southernmost position in the boreal winter (Fig. 4a, December–February), in
agreement with the observations of King et al. (2013) based on the measurements of the MODIS sounders on board the Terra
and Aqua satellites. In the tropical regions (up to about 25° N and S), the seasonal variations in Africa South and North of the

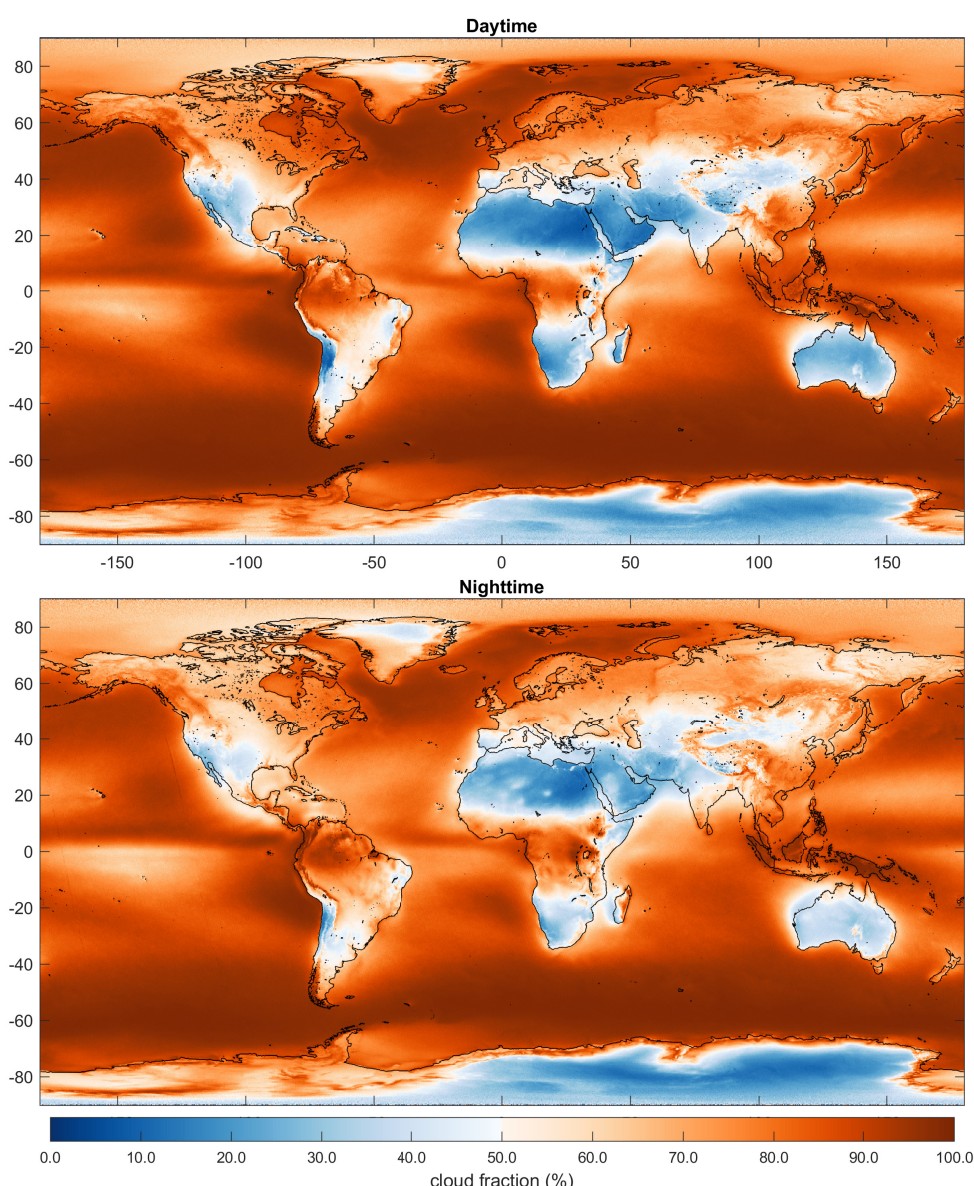

**Figure 3.** Mean cloud cover (%) calculated using the IASI neural network cloud detection algorithm between 2008 and 2020 separately for day (top) and night (bottom) observations in a $0.25° \times 0.25°$ grid.





equator, in Central South America, in India and in South East Asia are mainly associated with the monsoon with a maximum
cloud coverage observed during the boreal summer (winter) for the regions located North (South) of the Equator (Eastman
and Warren, 2010; Warren et al., 2015). Northern Australia also experiences cloudy conditions during boreal winter. At mid-
latitudes, in the Northern Hemisphere, minimum and maximum cloud amount are observed, as expected, during the summer
and the winter months, respectively.

At high latitudes, the Southern Ocean surrounding Antarctica exhibits high mean cloud amount, reaching up to 98% during
the boreal winter and with a minimum of about 88% during the summer. A more pronounced seasonal cycle is observed
around the Weddell sea and the Ross sea, with a decrease by an amplitude of about 25% observed during the boreal summer
(Fig. 4c, June–August). Over the Antarctic Plateau where the mean cloud amount is low, a clear seasonality is also observed.
The cloud fraction varies from about 16% in spring (Fig. 4b, March–May) to about 40% during autumn (Fig. 4d, September–
November). Western Antarctica, for its part, is more cloudy with a peak of 72% observed in the boreal winter and a minimum
of 52% in spring. Similar seasonal features and cloud occurence at high Southern latitudes have also been reported in previous
studies relying on CALIPSO observations (Adhikari et al., 2012; Listowski et al., 2019; Karlsson and Devasthale, 2018). In
the Northern Hemisphere, the Arctic region shows a strong seasonal cycle as well, more pronounced for the higher latitudes,
with a maximum during the boreal summer and a minimum in the winter. These seasonal patterns are in agreement with the
observations derived from CALIPSO in Karlsson and Devasthale (2018) and with those reported in Eastman and Warren (2010)
based on land, ships and drifting sea ice weather stations.

## 4 Assessment and intercomparison

### 4.1 Climatological mean

To give a first assessment of the IASI-NN cloud product, we compare here the NN cloud mask with other existing cloud
products. We start with average cloud distributions. In total, seven different products are selected for the comparison: the oper-
ational IASI-L2, the CIRS-LMD also directly based on the IASI measurements, the L1C-AVHRR and three cloud climate data
records (CDRs) built from the measurements of the AVHRR sounder on board the Metop-A satellite. Those are the CLARA-
A2.1 produced by the EUMETSAT Climate Monitoring Satellite Application Facility (CM-SAF) (Karlsson et al., 2013, 2017),
the Level-3U ESA Cloud_cci (Stengel et al., 2017) and the PATMOS-x (Pathfinder Atmosphere Extended) (Heidinger and
Pavolonis, 2009; Heidinger et al., 2014) datasets. We also include a comparison with the Aqua/AIRS L2 (AIRS+AMSU) V7.0
product, despite the different overpass time with IASI (01.30 AM and PM; AIRS Project, 2020). A short description of each
cloud product is provided in Table 2. The IASI-L2, the CIRS-LMD and the L1C-AVHRR are also briefly described in the
introduction. The three CDRs are based on different retrieval approaches (see Table 2 and references therein). In addition to a
cloud mask, they also include a full cloud characterization. The data are provided at a resampled resolution of $0.05° \times 0.05°$
grid for the CLARA-A2.1 and the ESA Cloud_cci and at $0.1° \times 0.1°$ for PATMOS-x. The AIRS-AMSU L2 product integrates
cloud top level information and the cloud effective fraction.

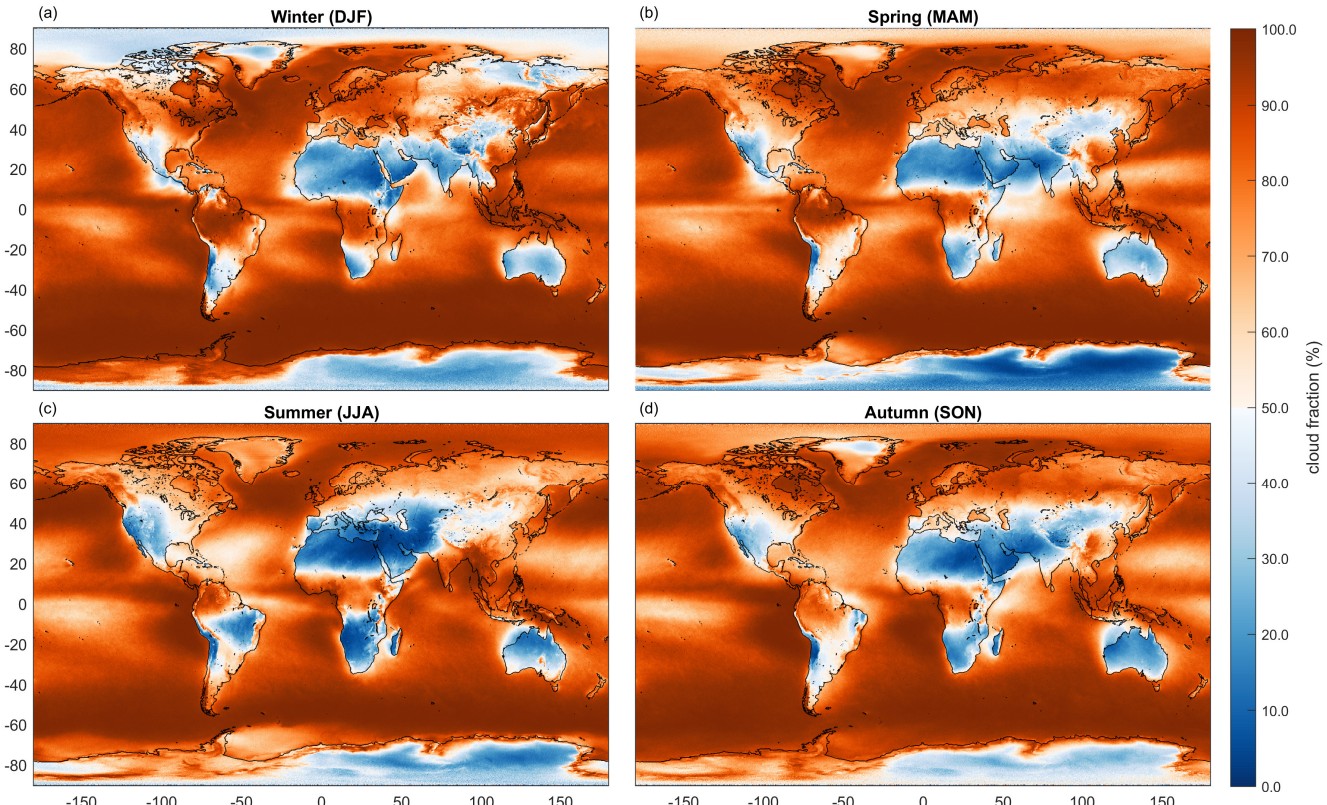

**Figure 4.** Seasonal mean daytime cloud cover (%) calculated from the IASI neural network cloud detection algorithm between 2008 and 2020. Top left: Winter (December–February), top right: Spring (March–May), bottom left: Summer (June–August), bottom right: Autumn (September–November.

The intercomparison is performed over the 2016 data, except for the IASI-L2 product which is derived from 2020 and the AIRS-AMSU products from 2015 (as AMSU failed in September 2016). For the IASI-L2, this choice was made so that the comparison is performed against the latest version of the L2 which was used for the training of the NN. For the IASI-L2, the L1C-AVHRR and the AIRS-AMSU products, as a cloud mask is not provided, we considered as clear the scenes with a cloud fraction in the field-of-view (FOV) strictly equal to 0%.

Fig. 5 shows the global mean cloud cover (%, daytime measurements only) in a $0.25° \times 0.25°$ grid for the IASI-NN and the seven cloud products considered for the inter-comparison. We also included the mean cloud amount distribution derived from the NN but with the threshold doubled over seas for the separation between clear-sky and cloud scenes (threshold of 0.55 instead of 0.275). For the three cloud CDRs, their initial resolution was degraded to match the 0.25° grid. The cloud fraction is calculated as in Sect. 3 (i.e. the number of observations flagged as cloudy over the total number of observations). As expected, the IASI-NN and the IASI-L2 products (Fig. 5a and c) are very similar with an identical mean cloud amount of 75.7% over the whole globe, a correlation coefficient and a mean of the absolute difference between the two distributions of 0.91 and 6.5%,



**Table 2.** Overview of the cloud characterization and retrieval algorithm for the datasets considered in this study. List of acronyms: CMA = cloud mask, CTP = cloud top pressure, CTMP = cloud temperature, CA = cloud amount, CPH = cloud phase, CE = cloud emissivity, COD = cloud optical depth, $r_e$ = cloud effective radius, CTO = cloud top level, CWP = cloud water path

| Product | Instrument(s) | Platform(s) | Algorithm | Retrieved parameter | References |
|---|---|---|---|---|---|
| Operational IASI-L2 | IASI | Metop | Until v6.4: (1) Cloud detection: - AVHRR collocated CMA, - NWP, - NN on IASI and AVHRR measurements (2) Characterization: $CO_2$-slicing and $\chi^2$ method. Since v6.5: - Cloud fraction: Optimal Estimation - Cloud detection derived from the retrieved cloud fraction | CMA, CTP, CTMP, CA, CPH | August et al. (2012), IASI Level 2: Product Guide. |
| LMD-CIRS | HIRS, AIRS, IASI | NOAA, Aqua, Metop | Weighted $\chi^2$ method (channels around 15 $\mu$m) | CTP, CTMP, CE, cloud type (8 in total, one for clear-sky) | Stubenrauch et al. (1999), Feofilov and Stubenrauch (2017), Stubenrauch et al. (2017). |
| IASI NN | IASI | Metop | Supervised NN (input: 45 IASI channels) | CMA | This work. |
| L1C-AVHRR | AVHRR | Metop | Sequence of threshold tests based on BT and inter-channel differences in the IR, vis and NIR + NWP forecast data | CA | August et al. (2012), Section 5.4.4 of the "EPS Ground Segment AVHRR L1 Product Generation Specification", EUM.EPS.SYS.SPE.990004. |
| L3U AVHRR-AM (ESA Cloud_cci) | AVHRR | NOAA-12, 15, 17 Metop | CC4CL v3.0 retrieval system: (1) Cloud detection: ANN using the AVHRR channel radiance, illumination, scan angles and auxillary data (2) Cloud typing: Threshold decision tree (3) Characterization: Optimal Estimation | CMA, CTP, COD, $r_e$, Cloud type (gridded $0.05^\circ \times 0.05^\circ$) | Stengel et al. (2017). |
| CLARA-A2.1 (CM-SAF) | AVHRR | NOAA, Metop | A. NWC SAF PPS cloud software: (1) CMA and CA: Multispectral thresholding technique, (2) CTO: - Comparison simulated and measured radiances, - inter-channel BT differences, B. CPP algorithm: CPH, COD, $r_e$, CWP: LUT approach | CMA, CA, CTO (T, P, Height), CPH, COD, $r_e$, CWP, + Joint cloud histogram + Surface radiation budget + Surface albedo (gridded $0.05^\circ \times 0.05^\circ$) | Karlsson et al. (2013), Karlsson et al. (2017). |
| PATMOS-x | AVHRR | NOAA, Metop | (1) Cloud detection: Bayesian classifiers derived from CALIPSO, (2) Characterization: Optimal Estimation | CMA, CTP, CE, COD, $r_e$, CWP (gridded $0.1^\circ \times 0.1^\circ$) | Heidinger and Pavolonis (2009), Heidinger et al. (2014). |
| AIRS-AMSU L2 v7 | AIRS + AMSU | Aqua | Comparison between observed and calculated cloud-cleared AIRS radiances | CTO (T, P, Height), Cloud effective fraction | Susskind et al. (2003), Kahn et al. (2014), AIRS Project (2020). |

respectively (5% when the comparison is done against the IASI-NN distribution for 2020). The same regions of high/low cloud load appear. In general, though, the IASI-NN mean cloud fraction is slightly lower (of about 6%) than the IASI-L2 in the

intertropical regions (between 20° N and S) while the opposite is observed for the mid- and high latitudes North and South of the Equator. This is particularly visible in the Pacific, south of the ITCZ and in the Southern Pacific and Southern Atlantic. Over Antarctica, while the cloud fraction is close on average (within 2%), large regional disparities are observed especially in the Eastern part. Along the Antarctic coasts, the cloud distribution derived from the NN is generally higher than the L2 up to about 40%. Over the Antarctic Plateau, in contrast, the cloud cover is about 5% lower for the NN (but up to 20-25% locally).

Note that the differences between the L2 and the NN do not allow to conclude to a better performance in the cloud detection for one or the other product as there is no guarantee that the cloud attribution of the L2 is always correct.

The second IASI-derived cloud product, the CIRS-LMD (Fig. 5d), appears much less conservative in the clear-sky attribution with a mean global cloud fraction (land and sea observation togheter) of about 66% (74% above oceans). Besides that, the same cloud patterns of low and high cloud coverage appear, as reflected by the very good correlation with the NN cloud product





(r = 0.88). The correspondence in the cloud fraction with the NN becomes much better when the threshold considered for separating the clear-sky from the cloud scenes over oceans is doubled (Fig. 5b). With this parametrization, the mean cloud coverage reaches 69% globally (76% above oceans only). Note also for both the L2 and the CIRS-LMD product the similar features that were reported for the IASI-NN product (see Sect. 3) in the regions of tall mountain ranges associated with a weaker sensitivity to cloud detection.

The four AVHRR-derived cloud products, for their part, show very different mean cloud amounts. The more conservative dataset in the cloud detection is the L1C-AVHRR product (Fig. 5e). Over seas, for tropical and mid-latitude regions (between 50° N and S), the cloud amount is close to the one from the IASI-NN product (mean of the absolute difference of about 3%). At higher latitudes, the difference increase to 9%. Above land, the differences are much more pronounced and increase with latitude (from about 20% in the tropics and mid-latitudes to 32% above 50° N and S). This is particularly evident above

Antarctica where a large overestimation of the cloud fraction is observed for the L1C-AVHRR product (mean cloud coverage of 96%). Large differences that are traceable to specific emissivity features are also present above deserts and over high mountains (e.g. over the Sahara).

  The three other AVHRR products (CLARA-A2, CCI and PATMOS-x, Fig. 5g,h,i) show cloud amounts close to those from the CIRS-LMD and the IASI-NN product with doubled threshold for the tropical and mid-latitudes regions (around 60% on

average between 50° N and S) but with lower fractions observed over the subtropical gyres. The main difference between the three products lies in their sensitivity to cloud detection at high latitudes, especially above 75° N and over Antarctica. For the latter, the CLARA-A2 and CCI products are well below the three IASI-derived cloud products (mean cloud coverage of 30% versus 48% for IASI-NN) and are likely underestimating the cloud amount (Karlsson and Devasthale, 2018). The PATMOS-x product, in contrast, is much more strict in the cloud free scenes attribution with a mean cloud fraction over Antarctica of 87%.

Finally, the more conservative of the cloud products analyzed here is the AIRS-AMSU L2 (Fig. 5f). Globally, the mean cloud fraction for the year 2015 is 88%, about 12% higher than the NN-derived cloud product. Over seas, for the tropical and mid-latitude regions (between 50° N and S), the mean value even reaches 99% (85% for the NN product). Deserts also show a high cloud coverage compared to the other selected products, with for example a mean cloud fraction of 74% for the Sahara desert against 24% with the NN product. Only high latitudes present very similar cloud amount to the NN with an average

cloud coverage of 76% and 65% above and below 60° North and South, respectively.

## 4.2 Time series

Fig. 6 shows the time series of the global mean fraction of cloud free scenes for the NN, the EUMETSAT IASI-L2 and the CIRS-LMD cloud masks, derived from Metop-A and Metop-B between 2008 (2013 for Metop-B) and 2020. Currently, the CIRS-LMD data are only available until 2019. Results are presented for land and sea observations together and separated.

Both the NN and the CIRS-LMD cloud product show an excellent stability over time with no visible trends and a very good consistency between the two Metop instruments (absolute difference lower than 1% on average for the NN). In contrast, the L2 cloud free time series reveal sharp discontinuities coinciding with version changes. In addition, a clear positive trend is also visible between 2012 and 2020, likely due to the $CO_2$ concentration increase in the atmosphere.



**Figure 5.** One year average cloud cover (%) from IASI, AIRS and AVHRR-derived cloud products (daytime observations). The year considered for the averaging is 2016, except for the EUMETSAT IASI-L2 product (2020) and the AIRS-L2 product (2015). The global mean cloud cover is provided on top of each panel.





As expected, the correspondence between the fraction of clear scenes from the NN and the last version of the L2 (version
6.5, from December 2019 onward) is excellent with differences lower than 1% on average for both land and sea measurements.
The fraction of clear scenes above sea and land are about 14% and 42% on average, respectively. The CIRS-LMD product, for
its part, appears to be less conservative in the cloud detection over seas with about 25% of all scenes flagged as clear but show
comparable values (42%) over land.

When looking at the interannual variations, a clear seasonality appears above land for both the NN and the CIRS-LMD
products with a peak in the clear-sky scenes during the boreal summer. A good agreement in the seasonality is also found with
the L2 cloud product from the last version of the L2 but, oddly, the seasonality seems out of phase with the NN (peak occuring
during the boreal spring) between 2012 and 2019 (corresponding to the version 5.3 to the version 6.4 of the L2). The reason
for the differences has not been investigated further.

### 4.3   Dust detection

One important aspect to consider when assessing the quality of a cloud mask is its ability to distinguish clouds from dust
plumes. Fig. 7 shows examples of days (two in 2013 and two in 2020) where dust plumes are clearly visible above Africa or
the Eastern Atlantic (right column). Those were selected by analyzing the MODIS (Terra) corrected reflectance (true colors)
imagery (https://worldview.earthdata.nasa.gov) and a dust index developed by Clarisse et al. (2019) quantifying the strength
of the dust signal in the IASI spectrum. For each of them, we plotted the cloud flag derived from the L2 (left) and from the
NN (middle column). On top, we also displayed the contour plot of the dust index for two different levels (index of 10 and 20,
respectively).

As it has already been reported in Clarisse et al. (2019), older versions of the operational IASI-L2 cloud products are affected
by the presence of dust for the detection of cloud scenes. This can be seen for example on the 3 and the 7 of June 2013 (left
panel of the two first lines) with version 5.3.1 of the L2 where the regions of high dust loads (index higher than 10 and 20) are
often flagged as cloudy while no clouds are visible on the MODIS imagery.

In contrast, the NN and the last version of the L2 (v6.5) cloud products seem to fairly differenciate clouds from dust plumes,
as observed for example on the 8 of June 2020 (third line) and also in 2013 for the NN. However, in rare cases, the cloud
detection is still affected by the presence of dust, especially over land in the center of dust plumes. These false cloud detection
seem to occur more frequently in the L2 than in the NN. This is for example the case on the 7 of June 2020 (bottom panels)
where the dust index indicates the presence of a high amount of dust over Western Mauritania with no cloud visible on the
MODIS imagery. In the area, the NN (middle panel) correctly flag the IASI measurements as cloud-free while the L2 reports
the presence of clouds for most of the pixels. Similarly, on the 8 of June 2020, the bottom part of the area above Niger where a
high amount of dust is observed (dust index > 20) is flagged as cloudy in the L2 and as clear in the NN while the MODIS map
shows the presence of only a few sparse clouds (for the most part detected by the NN). The relatively better performance of the
NN compared to the L2 in correctly differentiating dusts from clouds may seem counterintuitive as the NN was trained to follow
the L2. However, as mentioned in Sect. 2, among the 10 different trainings performed, we selected the least affected with dust.
The choice was made by analyzing the cloud mask retrieved over 17 different dust storms in parallel with the corresponding

**Figure 6.** Time series of the daily mean fraction of cloud free scenes for the L2, NN and LMD cloud products (metop-A and metop-B) globally for land and sea (top panel) and for sea (middle) and land (bottom) observations only. The thick lines represent the 30-days moving average.



MODIS true colors imagery and the dust index (Clarisse et al., 2019). In general, on the global scale, the performance of the different trainings were very close. Note that, apart from the dust, a good agreement between the NN and the last version of the L2 is also found for the areas flagged as clear and cloudy and match with the MODIS true colors maps.

## 5 Conclusions

With already more than 15 years of continuous and consistent measurements and at least another 25 years to come with the recent launch of IASI on Metop-C and the future launch of IASI-NG (New Generation, with a spectral sampling of 0.125 cm$^{-1}$) on the Metop-SG suite of satellites (Crevoisier et al., 2014), the IASI dataset is becoming a reference fundamental climate data record. To be exploited to its full potential, it requires an accurate and unbiased cloud filter. While several cloud products of high quality already exist, they are however generally either not homogeneous on the whole IASI time series or not strict enough in the cloud detection for using in satellite data retrieval schemes of geophysical variables. In this paper, we have presented a new algorithm for the identification of cloud-free scenes in the IASI field-of-view that is well suited to study trends for IASI-derived trace gases and other climatological variables. The method relies on a neural network (NN) taking a set of 45 IASI radiance channels as input and trained with the last version (v6.5) of the operational IASI Level2 (L2) cloud product as reference. Usually, cloud detection and characterization methods exploit the information derived from $CO_2$ sensitive channels. Here, the channels were selected outside the regions of absorption of $CO_2$, $CH_4$, $N_2O$, CFC-11 and CFC-12 to avoid any long-term biases, as their concentrations are evolving over time in the atmosphere. Despite this, we managed to reach excellent performances for the training (87%) indicating that the information for detecting clouds is also present in other regions of the spectrum. We have shown that the retrieval is both sensitive to the cloud detection and consistent over the entire IASI lifespan and between the different copies of the IASI instrument on board the suite of Metop satellites. It also differentiate fairly well cloud from dust plumes. While the agreement with the IASI-L2 cloud product was generally very good with a correspondence of about 87% overall, differences have been reported over some regions, especially above seas over the subtropical gyres and in the Southern Pacific and the Southern Atlantic, but without being able to conclude to a better performance for either product. The agreement with other existing products was generally good in the main cloud regimes but with sometimes large differences in the mean cloud amount (up to 10% on average), especially with the three AVHRR cloud climate data records and with the CIRS-LMD IASI cloud product. Given its good performances, the IASI NN cloud product is planned to be implemented in the near future in the ANNI (Franco et al., 2018), the dust (Clarisse et al., 2019) and the spectrally resolved Outgoing Longwave Radiation (Whitburn et al., 2020) IASI retrieval frameworks. Note finally that, in case of future improvement in the IASI-L2 cloud product, the NN could easily be retrained and a complete reprocessing of the entire time series of the IASI NN cloud mask could be performed in a short time.

*Code availability.* The analysis codes can be made available upon request to the corresponding author.

**Figure 7.** Cloud detection over North-West Africa from the EUMETSAT IASI-L2 cloud product (left column) and the IASI-NN detection algorithm (middle column) for 4 days (2020/02/13, 2020/02/24, 2020/06/17, 2020/06/18) affected by dust plumes. Yellow and Blue circles refer to cloudy and cloud-free scenes, respectively. The right column shows the MODIS (Terra) corrected reflectance (true colors) imagery for the corresponding days (from NASA Worldview). The contours represent two levels of dust index (10 red and 20 black).



*Data availability.* The daily IASI cloud dataset will be made freely available for all users through the IASI-FT website: https://iasi-ft.eu/.

*Author contributions.* S.W. and L.C. conceptualized the product. S.W. developed the algorithm, performed the evaluation of the data and
350  drafted the paper. All authors contributed to the manuscript writing.

*Competing interests.* The authors declare that they have no conflict of interest.

*Acknowledgements.* IASI has been developed and built under the responsibility of the Centre National d'Études spatiales (CNES, France).
It is flown on board the Metop satellites as part of the EUMETSAT Polar System. The IASI L1C data are received through the EUMETCast
near-real-time data distribution service. This project has received funding from the European Research Council (ERC) under the European
355  Union's Horizon 2020 research and innovation program (grant agreement No 742909, IASI-FT advanced ERC grant). It was also supported
by the Prodex arrangement HIRS (Belspo-ESA). Simon Whitburn is grateful to the ERC for funding his research work. L. Clarisse is a
research associate (Chercheur Qualifié) supported by the Belgian F.R.S.-FNRS. C. Clerbaux is grateful to CNES for scientific collaboration
and financial support. We acknowledge the use of imagery from the NASA Worldview application (https://worldview.earthdata.nasa.gov),
part of the NASA Earth Observing System Data and Information System (EOSDIS).



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
