# Peer review of "A CO2-independent cloud mask from IASI radiances for climate applications"

_Atmospheric Measurement Techniques, 2022_

## Author Response (AR1)

**We would like to thank the reviewers very much for their constructive feedback on the paper and their useful comments and suggestions, which all have been answered or addressed and significantly improved the paper. Point-by-point responses are provided below. The original review comments are shown in black, our responses are shown in blue and bold. A "tracked changes" version of the manuscript is also appended.**

Reviewer 1:

This manuscript describes a new algorithm to discriminate cloudy versus cloud-free scenes in IASI pixels. It uses a neural network approach, trained to reproduce the current last version of the EUMETSAT IASI level 2 cloud product, and ends up indeed with similar results as that training product, with the benefit that it can easily be processed along the whole IASI time series and provide therefore a homogeneous cloud mask time series, while the EUMETSAT IASI cloud product undergoes a number of discontinuities during the whole IASI time period. This data will be useful for all other types of climate data records reprocessing based on IASI data, all requiring cloud-free scenes.

The paper is well organized, well written, well presented, and the scientific work is sound and well described. To my point of view, it will be suitable for publication after the following comments are addressed.

**Major comments**

1. The IASI-related bibliographic references should be made a bit larger than citing work from the co-authors and their teams. This is especially obvious in the introduction, lines 29 to 32, where discussing in general trace gas, aerosols and radiation retrieval/work (not even specifically from IASI) and citing only 3 papers, all from the team(s) authoring the current manuscript. I also quickly scanned the full bibliographic list: all cited papers containing the word "IASI" (except 2, one on IASI assimilation and one on AVHRR within IASI pixels) contain co-authors from the current manuscript. This feels weird.

**It is true that even if the paper describes a cloud mask algorithm specifically dedicated to the IASI measurements, we could have emphasized more the work from other teams working with other instruments, especially for the discussions not especially related to IASI. We have now added the following references to the manuscript:**

**Lines 27-33: Besides their importance for modeling the Earth's climate, accurate detection of cloud-free (clear) scenes from satellite measurements is also an essential preprocessing step for most climate and atmospheric satellite applications. This is especially the case in most trace gas (e.g. Warner et al., 2013; Van Damme et al., 2017) and dust retrieval schemes (e.g. DeSouza-Machado et al., 2010; Capelle et al., 2018; Clarisse et al., 2019) or to derive the Earth Outgoing Longwave Radiation (OLR) budget (e.g. Loeb et al., 2003; Chen and Huang, 2016; Whitburn et al., 2020), [...].**

**Concerning the remark on the predominance of the co-authors in the IASI cited papers, the reason is that the co-authors of this paper belong to three large teams (EUMETSAT, ULB/SQUARES, LATMOS/IPSL) that are deeply implicated in the development of the IASI retrieval algorithms since the beginning of the IASI/Metop mission (including all the official IASI L2 products).**

**References:**

- Warner, J., Carminati, F., Wei, Z., Lahoz, W., and J.-L. Attié (2013). Tropospheric carbone monoxide variability from AIRS under clear and cloudy conditions. *Atmos. Chem. Phys., 13*, 12469-12479, https://doi.org/10.5194/acp-13-12469-2013

- DeSouza-Machado, S.G., Strow, L. L., Imbiriba, B., McCann, K. K., Hoff, R. M. Hannon, S. E., & Torres, O. (2010). Infrared retrievals of dust using AIRS: Comparisons of optical depths and heights derived for a North African dust storm to other collocated EOS A-Train and surface observations. *Journal of Geophysical Research, 115*, D15201. https:///doi.org/10.1029/2009JD012842
- Capelle, V., Chédin, A., Pondrom, M., Crevoisier, C., Armante, R., Crepeau, L., & Scott, N. (2018). Infrared dust aerosol optical depth retrieved daily from IASI and comparison with AERONET over the period 2007–2016. *Remote Sensing of Environment, 206*, 15–32. https://doi.org/10.1016/j.rse.2017.12.008
- Chen, X., and X. Huang (2016). Deriving clear-sky longwave spectral flux from spaceborne hyperspectral radiance measurements: A case study with AIRS observations. *Atmos. Meas. Tech., 9*, 6013-6023, https://doi.org/10.5194/amt-9-6013-2016
- Loeb, N. G., N. M. Smith, S. Kato, W. F. Miller, S. K. Gupta, P. Minnis, and B. A. Wielicki (2003). Angular distribution models for top-of-atmosphere radiative flux estimation from the Clouds and the Earth's Radiant Energy System instrument on the Tropical Rainfall Measuring Mission satellite. Part I: *Methodology. J. Appl. Meteor., 42,* 240–265, https://doi.org/10.1175/1520-0450(2003)042<0240:ADMFTO>2.0.CO;2

2. Lines 54-56 Could you add a reference to support your statement (last sentence of the paragraph) and maybe mention what was the IASI-CIRS developed for, if not for cloud masking in satellite retrievals?

**We have adapted the manuscript as follows:**

**Lines 55-58:** However, the product is not strict enough to be used in the cloud removal preprocessing phase in satellite data retrieval schemes of geophysical variables. It was also never designed for this purpose but rather for establishing global climatologies of cloud properties (Stubenrauch et al., 2017).

3. Line 187: what are those "emissivity features associated with a lower sensitivity" and why does it only affect night-time data?

**The emissivity features observed in Figure 3 likely reflect inhomogeneities in the land surface properties. These particular cases, more visible at night because of the lower surface temperature, can easily be confused with clouds and are difficult to train with the neural network as they only represent a very small percentage of all the IASI measurements.**

Is this also present in the other IR cloud retrievals?

**Yes, as for the false cloud detections reported over the high mountain ranges, the emissivity features are also observed in the distribution from other cloud retrieval algorithms (even on daytime distributions, as seen on Figure 4).**

**We now mention this in the manuscript:**

**Lines 200-205:** [...] or to the heterogeneity of the topography within the IASI pixel which make the distinction between clear and cloudy pixels more difficult. In addition, for the nighttime distribution, patches of slightly higher cloud coverage are observed over deserts, in particular for the Sahara Desert, which probably reflect emissivity features due to inhomogeneity in the land surface properties associated with a lower sensitivity. These patterns over the high mountain ranges and deserts are also observed in the distributions derived from other cloud products (see Sect. 4).

4. For the intercomparison, I find a bit problematic to have all data from 2016 except two. I understand the reasons to have the IASI-L2 from 2020 but then why not all data for 2020? – for AMSU it is impossible, then obviously you should keep 2015, but then I would have a second plot for 2015 with AMSU and IASI NN. In any case, comparing yearly averages for different years does not make much sense, except if you can prove that those yearly averages do not change over time.

**The reason for choosing the year 2016 is that not all datasets are available until 2020 so far. For example, the CIRS-LMD and the CLARA-A2 products are available until 2019, and the ESA Cloud_cci until 2016. We believe it is fair to compare different years as we are not performing here a validation against other products but simply an intercomparison to illustrate the strengths and weaknesses of the existing products. Moreover, at global scale and on average, the variability in the cloud amount and in the cloud distribution is relatively limited for the main features that are discussed in the manuscript. This can be observed when comparing the daily averaged NN cloud distribution (2008-2020) (Figure 3) and the cloud distribution for 2016 on Figure 5a and the time series of the mean fraction of cloud free scenes for the NN on Fig. 6.**

5. In the intercomparison with IASI-L2: when comparing the year 2020, which was used for the training, could you specify which percentage of the full year data was used in the training, and if it is small enough that the comparison is not skewed towards being "perfect" because it was the training set?

**The training was performed with 54 000 L2 clear and 54 000 cloud scenes. These represent only a very small percentage of the total number of IASI observations over one year (more than 1.2 million data per day, so less than 0.025%).**

6. Is there any way to assess which algorithm makes a better job (IASI-L2 or IASI-NN) in the cases where they do not agree? For example, select a number of specific cases and compare with MODIS Terra as is done for the dust aerosols? That would be a great addition to the paper, I think, being able to discern if the NN makes a better job than its training data, or if it is "only" similarly good but with easy long time series consistent processing.

**It was relatively easy to evaluate the ability of the NN and the L2 cloud products to distinguish clouds from dust by comparing with MODIS images because we could select large and homogeneous dust plumes transported over apparent cloud free regions (also relying on a dust index developed for IASI measurements). However, apart from these very specific cases, it would be very difficult to determine which product works better based on the comparison with MODIS. Cloud amount and cloud cover can evolve rapidly (due to transport and/or evaporation), especially in the case of thin cirrus or sparse clouds. Given the difference in the overpass time (9.30 vs 10.30) between IASI and MODIS and that the main differences observed between the L2 and the NN are generally found on the cloud edges, it would therefore be very complicated to draw conclusions.**

**In our opinion, based on the comparison we have done, the NN is doing as good as the L2 in most of the cases (which was the goal of the proposed algorithm) but seems to perform slightly better at differentiate cloud from dust plumes. This is mentioned in the manuscript at lines 245-246.**

**We have clarified the limits of the comparison between MODIS and IASI for identifying cloud scenes:**

**Lines 160-165:** […] The MODIS Terra corrected reflectance imagery for the same day is shown as well. An excellent correspondence is found for the large structures of high opaque clouds and the cloud-free regions (e.g. the North and South Africa, the Arabian Peninsula, …). For the regions characterized by

sparse cumulus or thin cirrus clouds, the comparison is more difficult because of the different overpass time of the two instruments (10.30 AM/PM for MODIS Terra and 9.30 AM/PM for IASI) as the spatial distribution of clouds can evolve very rapidly (due to evaporation, precipitation and transport).

7.  In the comparisons, the NN with the double threshold compares better to some other products. There is no discussion afterwards which threshold should be retained for the IASI-NN mask. I guess it is the "non doubled", but I think it is worth a sentence or two.

**Indeed, to be used as cloud removal preprocessing phase in satellite data retrieval schemes, we recommend using the 'non-doubled' threshold as this is the one that matches the best the IASI L2 cloud product which was used as a reference dataset. However, for other applications the user is free to set the thresholds that seem the most appropriate. We have clarified this in the manuscript:**

**Line 146-150:** [...] A separate threshold, of respectively 0.175 (± 0.020) and 0.275 (± 0.015), was defined for land and for sea measurements. Those are recommended when the NN cloud product is used for cloud removal preprocessing phase in satellite data retrieval schemes. However, as we demonstrate in Sect. 4.1, this threshold can be adjusted depending on the application. The uncertainty on the thresholds is estimated by evaluating the change in the threshold for a 1% increase in the difference between the L2 and the NN.

8.  In the time series, it is clear that the IASI-L2 sea undergoes a "drop" at the launch of v6.5; however, I am surprised to see that the CIRS-LMD matches quite nicely the IASI-L2 before v6.5 from 2013 onwards (before that, anyone would agree that the IASI-L2 has issues); so has it been proved that the latest IASI-L2 yields better results than the prior versions? If not, maybe the IASI-NN should be trained with the prior IASI-L2 version 6.4? And if yes, maybe the CIRS-LMD team could be consulted to try to understand why their (independent) data matches better the previous IASI version?

**Validation against CALIOP data is part of a routine monitoring of the L2 cloud product since 2019. Public monitoring reports are available at https://www.eumetsat.int/iasi-level-2-geophysical-products-monitoring-reports. The long-term statistics are updated each month, by appended new matchups into the evaluation. The current version of the cloud product is the result of many years of improvement to reach a quite mature state today. As it can be seen on Fig. 2.3 of the last report (August 2022), the release of the version 6.5 has brought a significant improvement in the cloud detection against version 6.4. Hanssen Kuiper's skill score and the Percent Correct comparing the L2 cloud mask and CALIOP have increased of about 35% and 15%, respectively (to reach about 75% and 87%). This has been emphasized in the manuscript:**

**Lines 96-102:** The goal of the proposed algorithm is to produce a sensitive and consistent (unbiased) cloud mask over the entire IASI lifespan using as a reference dataset the version 6.5 (v6.5) of the operational IASI L2 cloud product (August et al., 2012). The latter shows a clear improvement over the previous version (v6.4) when compared to the cloud products from the Cloud-Aerosol LIdar with Orthogonal Polarization (CALIOP) onboard the Cloud-Aerosol LiDAR and Infrared Pathfinder Satellite Observation (CALIPSO, Winker et al., 2007), as reported in the CALIOP-CALIPSO IASI Level 2 geophysical products monitoring reports available at https://www.eumetsat.int/iasi-level-2-geophysical-products-monitoring-reports (accessed online on August 10, 2022). The retrieval method presented here is based on a supervised NN relying on the IASI radiance spectra only, [...].

9.  About the specific "dust training set" (17 selected dust storms), could you confirm that the events taken to analyse the performance, in Fig 7, were not part of these 17 events used to select the best NN training with respect to dust?

**We think there might be a misunderstanding here. There is no specific dust training set. The training set was composed of a large set of clear and cloud scenes (according to the IASI L2 cloud product) taken randomly over one year (2020). The best training was then selected by, among others, analyzing its performance over 17 dust plume events. The events used to illustrate the performance of the IASI NN cloud mask over dust plumes (Fig. 7) are taken from these set of 17 dust storms. Note however that none of the observations of these 17 events were used for the training.**

10. In data availability, please be more specific about which data will be available: the cloud mask, or the output of the NN (and then the user has to decide of the threshold, as line 137 mentions that "this threshold can be adjusted depending on the application" – but then, except the double sea threshold for comparisons, no threshold depending on application is ever discussed).

**This has been clarified in the data availability section:**

**Line 368-370:** The daily IASI NN outputs will be made freely available for all users through the IASI-FT website: https://iasi-ft.eu/. For using as cloud removal preprocessing phase in satellite retrieval, we recommend adopting the thresholds mentioned in the manuscript for land and sea observations, but these can also be adjusted depending on the application.

**Typos and small corrections**

**Thank you for pointing out the typos and small corrections. All have been corrected.**

- Line 37 – at 9.30 AM and PM -> maybe add that this is local solar time and not UTC?
- Line 75 "algorithm that retrieveS" (s is missing)
- Figures 2 and 7: could you add the lat/lon as in Fig 3?

**For these two figures, we decided not to show the geographic coordinates as they were not helping for the interpretation and were reducing their effective width of the figures.**

- Line 152 providing a computational time without the technical information on the machine is a bit weird; I understand that the goal is to show it is pretty fast, without having to describe technically a system, but you could just mention "on a typical personal computer" or "on a type xxx HPC machine"
- Line 166 exhibit (remove the s)
- Figure 4 caption: missing the last )
- Table 2, first line: could you provide the start date of the version 6.5?
- Line 248 typo in "together"
- Figure 6: the y axis title says it is in %, while the numbers are clearly not in %
- Figure 7: the days in the caption do not match the days written on the plots; maybe also mention it is day-time data (9:30 AM)?
- Line 315: "dust" (without the s) or "dust aerosols"
- Line 406 (biblio) the IASI l2 PG citation would need some kind of number version, DOI, or weblink and access date, I think.

**Thoughts and suggestions – entirely left to your appreciation**

1. The title is slightly misleading (at least to me), as if maybe the cloud mask worked only in the absence of CO2 (which makes no sense), or does not contain CO2 (again, not much sense); I would suggest maybe using something like "CO2-independent" instead of "free"?

**This is indeed a good suggestion. Thank you.**

2. In the cloud mask plots, I find it counter-intuitive to have the clouds in yellow, especially when discussing the dust (which will usually call the colour yellow/orange in people's minds). Obviously, white would not be a good choice, but what about some kind of grey?

**We had the same feeling, but we have tried many color combinations and none of them were satisfactory (in terms of contrast, because the cloudy pixels are much more numerous than the clear-sky ones). The advantage here is that the pixels with and without clouds are clearly identifiable (even if the choice of the color is a bit counter-intuitive).**

3. Lines 156 to 159 why have 3 different words for the same concept? It is good to mention they mean the same thing, but I think it would be even better to use consistently the same term.

**We think that using the three terms helps to lighten the text for the reader.**

4. Often, I find the figures come a bit late with respect to the text addressing them, but I guess this is anyway dealt with during the final preparation by the journal crew.

**This is true, we will take care that the figures are properly placed in the final revision before publication.**

**Reviewer 2:**

The reviewed manuscript presents a cloud detection algorithm for the hyperspectral spaceborne IASI instrument(s). The method uses a neural network approach and uses only IASI radiances as an input. The authors paid specific attention to avoid the channels affected by trace gases with the concentration variable over time. When applied to a series of IASI measurements, the method shows physical results and its robustness is demonstrated over the whole period of all MetOp/IASI instruments' observations up to now.

The research is topical, the methodology presented in the article is sound, and the paper is well structured and written (except for some minor issues listed in "Technical corrections" section). Still, there are several issues I'd like to clarify/fix before recommending the manuscript for publication.

I have chosen "major revision", but the changes I suggest are easy to implement. I believe, if the authors add the suggested information to the manuscript, it will become irreproachable from the methodological point of view and it will have a broader impact.

**General comments:**

1. Even though the average cloud amount shown in Fig. 5 looks physical and reasonable, I see a general methodological issue in using IASI L2 cloud product for training the neural network. I do not question the quality of this product – as follows from lines 64–79 of the manuscript, the methodology is mature and the results are generally good, but there is one caveat. I would not hesitate if the training were based on some "ground truth" dataset coming from in situ measurements, ground- or space lidar, or some other instrument, but I see an inconsistency in using IASI itself as a reference, given that its time series (Fig. 6) shows certain artefacts in cloud cover. I do not ask the authors to redo the whole work, but it would be good to supplement it with some kind of validation of the training dataset using, e.g. CALIPSO clouds as a reference. To avoid the diurnal effects related to different overpass times, one can focus only on the clouds over the ocean. Perhaps, it would be sufficient to show several representative profiles of the training dataset and to compare them to overlapping CALIPSO cloud profiles (e.g. Chepfer et al., 2013). Or, better yet, show the 3-month average of the IASI L2 and CALIPSO (3 months are

required to get a full spatial coverage from CALIPSO). One can also use GEWEX Cloud Assessment files (https://climserv.ipsl.polytechnique.fr/gewexca/index-2.html) for the comparison, but in this case IASI L2 cloud product should be processed in accordance with GEWEX CA rules (Stubenrauch et al., 2013). Either way, this cross-validation of the training dataset seems necessary to wrap up the methodological part.

**Indeed, not a lot of details were given in the manuscript to justify the use of the version 6.5 of the IASI L2 cloud product as a reference, as also pointed out by the referee#1.**
**It is true that the IASI NN cloud product was not validated against CALIOP. However, the L2 cloud product is routinely monitored and validated against CALIOP data since 2019 and public monitoring reports are available at** https://www.eumetsat.int/iasi-level-2-geophysical-products-monitoring-reports**. The long-term statistics are updated each month, by appended new matchups into the evaluation. The current version of the L2 cloud product is the result of many years of improvement to reach a mature state today. As it can be seen on Fig. 2.3 of the last report (August 2022), the release of the version 6.5 has brought a significant improvement in the cloud detection against version 6.4. Hanssen Kuiper's skill score and the Percent Correct comparing the L2 cloud mask and CALIOP have increased of about 35% and 15%, respectively (to reach about 75% and 87%).**

**Here, the goal of the proposed algorithm is to produce an IASI cloud mask matching as best as possible the version 6.5 of the IASI L2 cloud product but consistent on the whole IASI time series (as the complete L2 cloud product has not been officially reprocessed yet with the latest version of the L2), and not affected by the changes in the atmospheric $CO_2$ (and the other long-lived species) concentrations. The good performance reached by the training and the other intercomparisons performed with the L2 cloud product show that the two products match very well and, therefore, the level of correspondence with CALIOP data should also be close to the one of the L2.**

**The manuscript has been updated to emphasize the good correspondence between CALIOP and the IASI L2 cloud product:**

**Lines 96-102:** The goal of the proposed algorithm is to produce a sensitive and consistent (unbiased) cloud mask over the entire IASI lifespan using as a reference dataset the version 6.5 (v6.5) of the operational IASI L2 cloud product (August et al., 2012). The latter shows a clear improvement over the previous version (v6.4) when compared to the cloud products from the Cloud-Aerosol LIdar with Orthogonal Polarization (CALIOP) onboard the Cloud-Aerosol LiDAR and Infrared Pathfinder Satellite Observation (CALIPSO, Winker et al., 2007), as reported in the CALIOP-CALIPSO IASI Level 2 geophysical products monitoring reports available at https://www.eumetsat.int/iasi-level-2-geophysical-products-monitoring-reports (accessed online on August 10, 2022). The retrieval method presented here is based on a supervised NN relying on the IASI radiance spectra only, [...].

2. As far as I understand, the present requirements to articles published in EGU open access journals include the distribution of the codes and/or the data used in the article. To my knowledge, the trained neural network of a kind applied in the manuscript can be represented by a couple of pages of ASCII-text in pseudocode (variables of the first layer = linear combination of the variables of zero-layer, second layer =…, …, result= linear combination of the variables of N-1th layer). It can be added to the manuscript itself or be provided as a supplement, but it should be certainly doable and it will be useful for the community.

**This is indeed a good suggestion. We have generated a MATLAB function (provided in ASCII-text format) allowing to easily run the network on MATLAB and containing all the training variables. An example code with one input (45 IASI channels and the orography) is provided as well.**

3. The contribution functions for the channels centered at the same or close wavelengths for IASI and AIRS should be close to each other, see e.g. (Feofilov and Stubenrauch, 2017) mentioned in the manuscript. Correspondingly, I'm almost sure that the neural network trained for IASI will be applicable to AIRS and maybe even to other instruments like HIRS. It would be good to apply the NN to AIRS L1 data to show the potential and versatility of the method. Just a single map in the appendix would generalize the approach and the NN file explained in comment #2 will enable the other researchers to calculate their own cloud mask on the fly.

**Indeed, this would have been a great addition to the paper. Unfortunately, not all the wavelengths used in the neural network to detect the presence of clouds are covered in the AIRS spectra (in particular, 13 channels around 2100 $cm^{-1}$ fall in the gap present in the AIRS spectrum between 1650 $cm^{-1}$ and 2175 $cm^{-1}$). It would of course be possible to develop an algorithm for the AIRS measurements using the same approach but it would then require to select a complete new set of AIRS channels and to train a new neural network. While this would be interesting, it is clearly out of the scope of this paper.**

**Specific comments:**

1. Lines 105–120: this text could be significantly simplified if Fig. 1 were supplemented with the vertical contribution functions (averaging kernels) for each channel. One can put these curves side by side, on top or at the bottom of Fig. 1. The actual scale is not that crucial, but the center and the halfwidth of the averaging kernel should be clearly visible. This is an important methodological point, which would explain the information content of the signals used in the approach. The NN itself has enough "black box" features, so anything that could be clarified should be clarified.

**It is true that by taking spectral regions affected by $H_2O$, $O_3$ and CO, the selected channels cover a range of altitude of vertical sensitivity comprised between 0 and about 30 km in clear sky conditions. However, the focus is on the detection of cloud scenes and the vertical contribution functions will have completely different shapes in the presence of clouds. Given this, we therefore prefer not to include the vertical contribution functions in the Figure 1 as we think it can be misleading. Nevertheless, as many of the input channels are located in the atmospheric windows, we can confidently assume that the network will be sensitive to clouds at any altitudes since the atmosphere at these wavelengths is transparent in the absence of clouds. We have clarified this in the manuscript:**

**Lines 123-125:** […] Note that, as most of the selected channels are located in the atmospheric window regions, the network should be sensitive to clouds at any altitudes since the atmosphere at these wavelengths is transparent in the absence of clouds.

2. Lines 130–132: I'm not sure I've got the idea here. Normally, the thresholds of this kind should be selected basing on the minimization of an error (or maximizing of the correlation coefficient) for two datasets. That's what is written in lines 134–136 below, and I agree with this approach. I'd suggest to leave only this part in this paragraph since the beginning is misleading.

**Indeed, the selected threshold is the one that minimize the difference between the IASI NN and the L2-derived cloud amount. The first sentence (Lines 130-132) was added only to emphasize that this threshold would be equal (or close) to 0.5 if the number of cloud-free and cloud scenes were equivalent on Earth, as the neural network was trained with half of the training set containing clouds and the other half without. We think this is interesting to be mentioned.**

It would be also interesting to have a look at the difference curve mentioned in line 135 to estimate the uncertainty of the threshold, but the authors can just do it themselves and provide a ±value along with the threshold.

**This is indeed a good suggestion. We have estimated an uncertainty corresponding to an increase of 1% in the difference between the L2 and the NN for land and sea observations separately. We have adapted the manuscript to include this:**

**Lines 146-150:** A separate threshold, of respectively 0.175 (± 0.020) and 0.275 (± 0.015), was defined for land and for sea measurements. Those are recommended when the NN cloud product is used for cloud removal preprocessing phase in satellite data retrieval schemes. However, as we demonstrate in Sect. 4.1, this threshold can be adjusted depending on the application. The uncertainty on the thresholds is estimated by evaluating the change in the threshold for a 1% increase in the difference between the L2 and the NN.

3. Fig. 2: In the left-hand side panel, I do not see the cloud structures of the right-hand side one. There are a lot of yellow circles in a cloud-free (?) area in the lower left section of the image. Perhaps, the light "haze" which one can see in the right-hand side corresponds to a real cloud, but it is not clear from the image. What is the correlation coefficient for these two panels and what is the r.m.s. of their difference?

**The Figure 2 was given as an illustration more than for validation purposes. In fact, cloud coverage comparison with MODIS (Terra) is only possible for large cloud structures. In the region that you mentioned, the sky seems characterized that day by thin cirrus and sparse cumulus clouds. Because of the high sensitivity of the IASI NN cloud mask algorithm, an IASI field-of-view will be declared as cloudy even if only a small fraction of the pixel is occupied by a cloud or in the presence of a very thin cloud. In these specific cases, the comparison between MODIS and IASI is very difficult as the two instruments have different overpass times (10h30 vs 9h30) and clouds distributions can evolve rapidly (transport, precipitation, evaporation). We have adapted the manuscript to clarify this:**

**Lines 160-165:** […] The MODIS Terra corrected reflectance imagery for the same day is shown as well. An excellent correspondence is found for the large structures of high opaque clouds and the cloud-free regions (e.g. the North and South Africa, the Arabian Peninsula, …). For the regions characterized by sparse cumulus or thin cirrus clouds, the comparison is more difficult because of the different overpass time of the two instruments (10.30 AM/PM for MODIS Terra and 9.30 AM/PM for IASI) as the spatial distribution of clouds can evolve very rapidly (due to evaporation, precipitation and transport).

4. Lines 175−180 : I wonder if the NN training with CALIPSO would improve the agreement.

**The agreement would potentially be better if CALIOP was used as the reference dataset. However, it would not be possible to develop a global cloud mask product using CALIOP as collocations with IASI are only possible at high latitudes because of the different overpass time (about 4 hours in the tropics) of Metop and CALIPSO (CALIOP-CALIPSO IASI Level 2 geophysical products monitoring reports, available at https://www.eumetsat.int/iasi-level-2-geophysical-products-monitoring-reports).**

5. Lines 280−289 : as in general comment #1, I stumble here because there is a certain issue in the source dataset used for training, and at the same time we have a NN based on this dataset, which doesn't have this issue. I understand that this is possible, but one has to discuss this inconsistency because methodologically the neural network is not supposed to be different from the training dataset.

**We think there might be a misunderstanding here. The discontinuities in the IASI L2 time series are the result of version changes in the retrieval algorithm (as the entire time series has not officially been reprocessed yet). As we mention in the manuscript (line 95), the neural network was trained only with output data from the version 6.5 of the IASI L2 cloud product (year 2020, which shows an excellent match with the NN in Fig. 6) and it was therefore expected that we obtain a consistent time series for the whole IASI period. To avoid a spurious trends over time, we also took care to avoid, for the input parameters of the NN, any channel that is affected by the absorption of changing greenhouse gases (lines 97-100). As the period considered for the training is short (one year), the existence of a trend in the L2 due to $CO_2$ concentration changes in the atmosphere is not affecting the training.**

6. Fig. 5: It is not clear whether the average cloud cover here was calculated considering "shrinking" of the lat/lon box when moving towards the poles. I made this exercise for Fig. 5d (see below) and I've got close values (67.0% and 66.9% for area-weighted and simple averages, respectively), but the differences might be larger for the instruments with better coverage of the polar areas. In any case, it is recommended to use the area-weighted values. Please, check.

**The average cloud cover provided were indeed simple averages. We have replaced them by the corresponding area weighted values. Differences are not very large (about 1-2%), except for the AIRS L2 (from 87% to 94%) and the PATMOS (from 75% to 69%) cloud coverage.**

7. Fig. 5: this is more a comment rather than an issue and I do not require the authors to squeeze another panel to Fig. 5, which is already busy, but I think that this plot is worth providing here. In Fig. 1 below, I show the mean CIRS-LMD IASI and mean CIRS-LMD AIRS for 2015. As one can see, they are quite similar because the channels of (almost) the same wavelength were processed with the same methodology. The remaining difference is due to diurnal variation (Feofilov and Stuberauch, 2019). What is important, the similarity of these plots indirectly proves the point made in general comment #3.

**Thank you for this very interesting remark. Unfortunately, as we explained in response to general comment 3, it is not possible to directly apply our network to the AIRS measurements because not all wavenumbers used in the NN are available in the AIRS spectrum.**

8. Line 305 : please, mark these areas on the maps in Fig. 7.

**We have updated the Figure 7 to clearly show the IASI pixels that are flagged as cloudy by the L2 and clear by the NN in presence of high dust loads (dust index > 10) and we have adapted the manuscript accordingly:**

**Lines 317-320:** [...] For each of them, we plotted the cloud flag derived from the L2 (left) and from the NN (middle column). On top, we also displayed the contour plot of the dust index for two different levels (index of 10 and 20, respectively). For the L2, the IASI pixels seen as cloudy by the L2 but as cloud free by the NN in presence of high dust loads (dust index higher than 10) are shown in pink.

**And Line 331:** In the area, the NN (middle panel) correctly flag the IASI measurements as cloud-free while the L2 reports the presence of clouds for most of the pixels (pink circles on Fig. 7).

[Figure]

*Figure 1 : Cloud detection over North-West Africa from the EUMETSAT IASI-L2 cloud product (left column) and the IASI-NN detection algorithm (middle column) for 4 days (2013/06/07, 2013/06/03, 2020/06/08, 2020/06/07, morning orbits) affected by dust plumes. Yellow and Blue circles refer to cloudy and cloud-free scenes, respectively. The right column shows the MODIS (Terra) corrected reflectance (true colors) imagery for the corresponding days (from NASA Worldview). The contours represent two levels of dust index (10 red and 20 black). The pink circles on the IASI L2 maps show the pixels flagged as cloudy by the L2 and cloud-free by the NN in presence of high dust loads (cloud index > 10).*

9. Lines 345–346 and 125-129: please, provide the information on training time and number of samples.

**The size of the data set is mentioned line 127 and the retrieval time for one day of IASI measurements is mentioned line 169-170. We have added a sentence about the training time in the manuscript, line 137-138:**

[...] In total, we performed 10 different trainings and we selected the least affected with dust (see Sect. 4.3). The training of the network takes, depending on the run, about 100-150 iterations and is completed in about 30 minutes on a typical personal computer. The performance of the selected training reaches 87.3% with an equivalent number of misdetections in the clear and the cloud group.

[Figure]

*Figure 1. Mean annual cloud amount retrieved by CIRS-LMD algorithm for (a) IASI-B (to be compared to Fig. 5d of the manuscript) and (b) AIRS (to be compared to Fig. 5f of the manuscript). The source data is the same as in (Feofilov and Stubenrauch, 2019).*

**Technical corrections:**

**Thank you for pointing out these technical mistakes. All have been corrected.**

- Line 18: please, change "on the weather" to "for the weather"
- Line 21: either "in detection … and in derivation" or "to detect … and to derive"
- Line 66: "performance" would be better here
- Line 75: "retrieves"
- Line 88: comma is missing after "In the next section"
- Lines 247, 255, 270, and elsewhere: the terms "more conservative" and "less conservative" require too much thinking, especially if clear sky attribution is mixed up with cloud fraction in one sentence. I would just write "assigns clear sky flags more (or less) often than …" to avoid any ambiguity.
- Lines 341–342: please, reformulate to make a distinction between AVHRR products and CIRS-LMD IASI product. In the current version, the sentence reads as if they are in the same category. As far as I understand from the manuscript, this is not the case.

**References used:**

- Chepfer H., G. Cesana, D. Winker, B. Getzewich, and M. Vaughan, 2013: Comparison of two different cloud climatologies derived from CALIOP Level 1 observations: the CALIPSO-ST and the CALIPSO-GOCCP, J. Atmos. Ocean. Tech., doi.10.1175/JTECH-D-12-00057.1
- Feofilov, A. G. and Stubenrauch, C. J.: Diurnal variation of high-level clouds from the synergy of AIRS and IASI space-borne infrared sounders, Atmos. Chem. Phys., 19, 13957–13972,
- Stubenrauch, C., and 22 co-authors, Assessment of Global Cloud Datasets from Satellites: Project and Database Initiated by the GEWEX Radiation Panel, Bull. Amer. Meteorol. Soc., 94(7), 1031-1049, doi:10.1175/BAMS-D-12-00117.1, 2013